# Is there a weekend effect in emergency surgery for colorectal carcinoma? Analysis from the German StuDoQ registry

**Friedrich Anger**[1]*, **Sven Lichthardt**[1], **Imme Haubitz**[1], **Johanna Wagner**[1], **Stefan Löb**[1], **Heinz Johannes Buhr**[2], **Christoph-Thomas Germer**[1,3], **Armin Wiegering**[1,3,4]*

**1** Department of General, Visceral, Transplantation, Vascular and Paediatric Surgery, University Hospital, Julius-Maximilians-University of Wuerzburg, Wuerzburg, Germany, **2** German Society for General and Visceral Surgery (DGAV), Berlin, Germany, **3** Comprehensive Cancer Centre Mainfranken, University Hospital, Julius-Maximilians-University of Wuerzburg, Wuerzburg, Germany, **4** Department of Biochemistry and Molecular Biology, Julius-Maximilians-University of Wuerzburg, Wuerzburg, Germany

\* anger_f@ukw.de (FA); wiegering_a@ukw.de (AW)

## Abstract

**Data Availability Statement:** The data that support the findings of this study are available from the DGAV StuDoQ registry but restrictions apply to the availability of these data, which were used under

### Background

Higher postoperative mortality has been observed among patients who received emergency colorectal surgery on the weekend compared to during the week. The aim of this study was to determine whether the weekday of emergency surgery affects the 30-day mortality and postoperative course in emergency colorectal surgery.

### Methods

Prospectively acquired data from the 2010–2017 German StuDoQ|Colorectal surgery registries were analysed. Differences in 30-day mortality, transfer and length of stay (MTL30) (primary endpoints), postoperative complications, length of stay and pathological results of resected specimens (secondary endpoints) were assessed. Multivariable analysis was performed to identify independent risk factors for postoperative outcome.

### Results

In total, 1,174 patients were included in the analysis. Major postoperative complications and the need for reoperation were observed more frequently for emergency colorectal surgery performed during the week compared to the weekend (23.01 vs. 15.28%, p = 0.036 and 17.96% vs. 11.11%, p = 0.040, respectively). In contrast, patients who received emergency surgery on the weekend presented with significantly higher UICC tumour stages (UICC III 44.06 vs. 34.15%, p = 0.020) compared to patients with emergency colorectal surgery on a weekday. Emergency surgery performed during the week was an independent risk factor for the development of severe postoperative complications (OR 1.69 [1.04–2.74], p = 0.033) and need for reoperation (OR 1.79 [1.02–3.05], p = 0.041) in the multivariable analysis.

license (ID StuDoQ-2017-0017) for the current study, and so are not publicly available. Contact information for data access request is: studoq@dgav.de.

**Funding:** The authors received no specific funding for this work.

## Conclusion

Emergency surgery for colorectal carcinoma in Germany is performed with equal postoperative MTL30 and mortality throughout the entire week. However, emergency surgery during the week seems to be associated with a higher rate of severe postoperative complications and reoperation.

## Introduction

There is an ongoing debate whether the patient outcome after emergency general surgery is influenced by the weekday on which it is performed [1–3]. In large retrospective studies, postoperative mortality was higher in patients receiving surgery on the weekend or national holidays, rather than on weekdays [4]. For specific surgical procedures, such as lysis of adhesions or partial colectomy, a higher postoperative mortality was observed for patients operated on the weekend [2]. However, patient data from different countries contradicted these results, showing no influence of the weekday on postoperative short- and long-term patient outcome [5, 6].

Regarding major colorectal resections, patient selection, rather than hospital variables seems to be the reason for a higher postoperative mortality in patients receiving surgery on the weekend compared to weekdays [3]. Factors that explain differences in risk adjusted mortality after general surgical emergencies in England have been identified, but the outcome measures relate to the day of admission not surgical treatment [7]. Of note, no difference in postoperative mortality by day of emergency general surgery has been observed [5, 6]. Data on the quality of surgical treatment and oncological outcome of the resected patients is missing. Furthermore, no information about the intra- and postoperative course of the patient is given. Despite an increase of postoperative mortality compared to elective colorectal surgery [2, 8], the overall and disease-free survival might not be affected in patients suffering from colorectal cancer [9].

The published data originates from different health care systems varying in the provision of acute and hospital care, weakening their comparability [10]. The aim of this study was to determine whether the day of emergency surgery for colorectal carcinoma influences the postoperative 30-day mortality, MTL30 status, surgical procedure and total number of resected lymph nodes, postoperative course and length of hospital stay in Germany on the basis of prospectively gathered data from the StuDoQ registry. The MTL30 is a validated endpoint parameter specific to the German health care system that combines mortality, transfer to higher level hospital, and length of stay beyond 30 days postoperative [11, 12].

## Methods

### Data source

Data was retrieved from the German StuDoQ registry, which was set up by the German Society for General and Visceral Surgery (DGAV) in 2010 to evaluate the quality of healthcare and risk factors for different types of surgery depending on the indicative disease. The StuDoQ| Colorectal surgery registry (www.dgav.de/studoq; www.en.studoq.de) is a prospective registry, which contains anonymized data of patients with colorectal diseases treated in German hospitals. Data from the participating clinics was included in a pseudonymized form and subjected to automatic plausibility controls. Validation by cross-checking with institutional medical controlling data is part of the annual certification process.

The DGAV established the publication guidelines (https://www.dgav.de/studoq/datenschutzkonzept-und-publikationsrichtlinien.html), while the Society for Technology, Methods, and Infrastructure for Networked Medical Research (http://www.tmf-ev.de/) established the data safety concept.

## Study population

All patients (n = 19,708), who were registered in the StuDoQ|Colorectal surgery registry after colorectal resection surgery between 2010 and 2017, were eligible for inclusion. Patients, who had received emergency surgery due to a complication of colorectal carcinoma, were included in the analysis. Resection surgery was classified as an emergency by the hospitals providing data to the SuDoQ registry. All surgeries performed on a weekend or nationwide holiday, were considered an emergency. Surgeries performed on holidays limited to certain federal states in Germany were excluded because the patient data set does not include information on the federal state. Patients with simultaneous resections of liver metastasis, endoscopic tumour resection, missing information on the type of surgery performed or location of the primary tumour were excluded from the study. Each individual patient was described by patient demographics (age, sex), clinical features such as the American Society of Anesthesiologists Classification (ASA-Score) and health status as well as histopathological factors (malignancy history, tumour location, histology, pTN-stage, number of resected lymph nodes), treatment details (surgical approach, operation time, conversion, anastomosis, deviation, mesocolic excision), postoperative course (surgical and non-surgical complications, graded by Clavien-Dindo), length of stay including readmission and logistic measures such as the day of surgery. Tumor stage was classified according to the TNM classification (2016 TNM8) of the American Joint Committee on Cancer (AJCC).

## Statistical analysis and outcome measures

The effect of the day of surgery was analysed in a two-step approach. First, patients were grouped by the day of emergency surgery to workday (Monday to Friday) or weekend (Saturday-Sunday / national holiday) procedures. Differences between groups in regard to patient characteristics and outcome parameters were assessed using Chi-square, Fisher exact, Kruskal Wallis or Mann Whitney $U$ test according to the data distribution and scale. Second, the association between the weekday of emergency colorectal surgery and the studied outcomes were adjusted for age (continuous variable), sex (male or female), ASA classification, operation time and UICC tumour stage in multivariable logistic regression. Statistical analyses were performed using IBM SPSS Statistics Version 24 for Windows (IBM Corp, Armonk, NY). A $p$ value $<0.05$ was considered statistically significant.

Data are presented as means with standard deviation (SD) for continuous variables, and as numbers with percentages for categorical variables. Primary outcome measures are the 30-day mortality and MTL30 status. Secondary endpoints consisted of overall postoperative morbidity and specific postoperative complications, length of hospital stay and pathological results of resected specimen. The total number of resected lymph nodes was compared as a nominal cut-off ($\geq$12 lymph nodes), based on the quality indicator of the German CRC audit (www.awmf.org) for the minimum number of resected lymph nodes.

## Results

### Patient characteristics

Of 1,174 patients included in this study, 1093 (93.10%) underwent open and 81 (6.90%) laparoscopic emergency surgery for colorectal cancer. 1030 (87.73%) patients received surgery

during the week (Monday-Friday), while 144 patients had emergency surgery on the weekend (Saturday-Sunday) or national holiday. 53.24% of the patients were male, mean age at the time of surgery was 71.1±11.6 years and mean BMI was 25.8±4.9 kg/m$^2$. Basic patient characteristics are summarized in S1 Table.

Patients, who underwent emergency colorectal surgery during the week, presented with significantly more coronary artery disease (20.4 vs 12.5%, p = 0.02), laparoscopic approaches (7.4 vs. 3.5%, p = 0.03), more extended hemicolectomies (12.3 vs. 8.3%, p = 0.002) and anterior rectum resections (10.4 vs. 2.8%, p = 0.002) compared to patients, who received emergency surgery on the weekend. There was no difference in diverting stoma placement between weekday and weekend surgeries performed. Laparoscopic surgery was performed significantly more often on a weekday (7.4 vs. 3.5%, p = 0.03), resulting in higher conversion rates in open surgery (9.7 vs. 5.6%, p = 0.03) (Table 1).

## Emergency surgery

Patients after emergent colorectal resections developed significantly more severe postoperative complications (grade IIIb or higher according to Clavien and Dindo) in case surgery had been performed on a week day rather than on the weekend (23.01 vs. 15.38%, p = 0.036). Consistently, there was a significant difference in the rate of re-operation after initial emergency surgery during the week (18.0%) compared to the weekend (11.1%) (p = 0.040). However, no difference could be detected regarding specific postoperative complications, namely anastomotic leakage, haemorrhage, ileus, fascial dehiscence or surgical site infections between the two groups. The length of hospital stay (17.3±12.1 vs. 15.8±8.8 days, p = 0.046), MTL30 status (19.4 vs. 16.1%, p = 0.37) and postoperative 30-day mortality (6.12 vs. 5.56%, p = 0.79) did not differ between the two patient groups, either (Table 2). The analysis of the pathological reports revealed a significant difference in number of positive lymph nodes resulting in significant higher N-stages in patients who underwent emergency surgery on a weekend (positive lymph nodes: 3.0±5.8 vs. 3.5±6.08, p = 0.0096; N1/2 52.53% vs. 65.38%, p = 0.013). This indicates higher tumour stages in colon carcinoma specimens after surgery on a weekend (UICC III 34.2 vs 44.1%, p = 0.020). No difference was observed in M- or local R-Status. Nodal count was chosen to assess for non-radical surgery (< 12 nodes per specimen was defined as the cut-off). The timing of emergency surgery did not influence the number of lymph nodes per specimen (Table 3).

## Multivariable analysis

In addition to the day of emergency surgery, several clinical as well as pathological factors were analyzed to determine risk factors for selected postoperative outcome measures. In the multivariable analysis, the ASA score was a strong predictor for the development of postoperative complications grade IIIB or higher by Clavien and Dindo (OR 1.88 [1.55–2.28], p<0.001), including reoperation (OR 1.43 [1.17–1.75], p<0.001). In addition, the ASA score was identified as an independent predictor of the MTL30 status (OR 2.51 [2.03–3.11], p<0.001) and 30-day mortality (OR 3.74 [2.57–5.44], p<0.001). UICC stage (OR 1.34 [1.00–1.80], p = 0.047) was associated with 30-day mortality after emergency colorectal resections, whereas the day of surgery was not. Emergency colorectal surgery on a business day was an independent prognostic risk factor for the development of major postoperative complications, defined as Clavien-Dindo Grade IIIB or higher (OR 1.69 [1.04–2.74], p = 0.033), and reoperation (OR 1.76 [1.02–3.04]. p = 0.041) (Table 4).

## Discussion

In the present study, we were able to show a higher postoperative morbidity for patients after emergent resection of colorectal cancer performed during the week compared to the weekend,

**Table 1. Patient characteristics by weekday of surgery.**

| Variable | Weekday | Weekend/Holiday | p-value |
|---|---|---|---|
|  | N = 1030 | N = 144 |  |
| Age [years], mean±SD | 71.3±12.6 | 70.6±13.2 | 0.62 |
| Sex |  |  |  |
| Male, n (%) | 544 (87.0) | 486 (88.52) |  |
| Female, n (%) | 81 (12.96) | 63 (11.48) | 0.44 |
| BMI [kg/m$^2$], mean±SD | 25.78±5.10 | 25.79±4.07 | 0.45 |
| ASA, n (%) |  |  |  |
| 1 | 79 (7.76) | 9 (6.25) |  |
| 2 | 405 (39.32) | 57 (39.58) |  |
| 3 | 451 (43.79) | 67 (46.53) |  |
| 4 | 91 (8.83) | 10 (6.94) |  |
| 5 | 4 (0.93) | 1 (0.69) | 0.77 |
| Functional status, n (%) |  |  |  |
| Independent | 836 (81.17) | 114 (79.17) |  |
| Partially dependent | 160 (15.53) | 24 (16.67) |  |
| Totally dependent | 34 (3.30) | 6 (4.17) | 0.81 |
| Comorbidities, n (%) |  |  |  |
| Arterial hypertonia | 596 (57.86) | 74 (51.39) | 0.14 |
| Coronary artery disease | 210 (20.39) | 18 (12.50) | **0.02** |
| Heart failure (NYHA I-IV) | 251 (24.37) | 31 (21.53) | 0.94 |
| Diabetes |  |  |  |
| • NIDDM | 132 (12.82) | 13 (9.03) |  |
| • IDDM | 66 (6.41) | 10 (6.94) | 0.40 |
| History of severe COPD | 60 (5.83) | 8 (5.56) | 0.90 |
| Chronic steroid use | 20 (1.94) | 3 (2.08) | 0.76 |
| Dialysis | 8 (0.78) | 1 (0.69) | 1.00 |
| Disseminated cancer | 129 (12.52) | 21 (14.58) | 0.50 |
| Weight loss (>10% bw) | 174 (16.91) | 19 (13.19) | 0.25 |
| Alcohol abuse | 60 (5.83) | 10 (6.94) | 0.60 |
| Liver cirrhosis | 28 (2.73) | 0 (0.00) | 0.04 |
| Surgical approach, n (%) |  |  |  |
| Open | 854 (82.91) | 131 (90.97) |  |
| Laparoscopic | 76 (7.38) | 5 (3.47) |  |
| Conversion | 100 (9.71) | 8 (5.56) | **0.03** |
| Total operation time [min], mean±SD | 156±61 | 156±58 | 0.77 |
| Resection strategy, n (%) |  |  |  |
| Right Hemicolectomy | 542 (52.62) | 83 (57.64) |  |
| Left Hemicolectomy | 254 (24.66) | 45 (31.25) |  |
| Extended Hemicolectomy | 127 (12.33) | 12 (8.33) |  |
| Anterior rectum resection | 107 (10.39) | 4 (2.78) | **0.002** |
| Diverting Stoma, n (%) |  |  |  |
| No | 801 (80.02) | 113 (80.14) |  |
| Yes | 200 (19.98) | 28 (19.86) | 0.97 |
| Time of stoma placement | 17 (1.70) | 0 (0.00) |  |
| Prior | 172 (17.18) | 25 (17.73) |  |
| Simultaneously After (re-operation) resection surgery | 11 (1.10) | 3 (2.13) | 0.40 |
| Anastomosis, n (%) |  |  |  |

*(Continued)*

**Table 1.** (Continued)

| Variable | Weekday | Weekend/Holiday | p-value |
|---|---|---|---|
| Hand-sewn | 433 (42.04) | 69 (47.92) | |
| Stapler | 597 (57.96) | 75 (52.08) | 0.18 |

although patients who underwent emergent colorectal resections on the weekend presented with higher tumour burden. Consequently, emergency colorectal surgery on a business day rather than on the weekend or a holiday was identified as an independent risk factor for the development of major postoperative complications and the need for reoperation. According to this data, there is no weekend effect for emergency colorectal cancer surgery in Germany.

Available literature on the weekend effect in general surgery is inconsistent, mostly based on heterogeneous patient cohorts, emergency surgery procedures, outcome measurements and data sources [1–3, 5]. As a consequence, results differ among studies and an explanation for the phenomenon is hard to identify [2]. Huijts et al. were the first to analyse the impact of emergency surgery on the weekend, specifically addressing colorectal cancer resections in the Netherlands [3]. In contrast to our study, they found a significant rise in postoperative mortality when the emergency colon cancer surgery was performed on the weekend. However, when looking at rectal cancer no significant difference in the 30-day mortality was observed, possibly due to the small sample size [3]. A similar distribution of emergent colon and rectal resections was observed in this study, as the great majority of patients received emergency resection surgery for colon cancer. Since tumour obstruction is the leading cause for emergent rectal resections [13], but decompressing stoma as bridge to surgery is the preferred route in German hospitals most of these patients might have received multimodality treatment prior to resection. When providing data to the StuDoQ registry, these patients are then classified as elective rectal resections and consequently missing in the underlying dataset of this study.

One of the leading hypotheses explaining a worse outcome after emergency surgery on the weekend is the limited availability of resources, including less experienced hospital staff [14].

**Table 2. Postoperative outcome by weekday of surgery.**

| Variable | Weekday | Weekend/Holiday | p-value |
|---|---|---|---|
| | N = 1030 | N = 144 | |
| Clavien Dindo grade $\geq$ IIIb, n (%) | 237 (23.01) | 22 (15.28) | **0.036** |
| Anastomotic leakage, n (%) | 78 (7.57) | 9 (6.25) | 0.34 |
| Hemorrhage, n (%) | 19 (1.84) | 4 (2.74) | 0.51 |
| Ileus, n (%) | 50 (4.85) | 5 (3.47) | 0.44 |
| Fascial dehiscence, n (%) | 89 (8.64) | 13 (9.03) | 0.88 |
| SSI, n (%) | | | |
| Superficial | 104 (10.11) | 14 (9.72) | |
| Deep | 40 (3.89) | 7 (4.86) | |
| Organ space | 22 (2.14) | 3 (2.08) | 0.96 |
| Pulmonary embolism, n (%) | 5 (0.49) | 1 (0.69) | 0.54 |
| Other, non-surgical complications, n (%) | 178 (17.28) | 30 (20.83) | 0.30 |
| Re-operation, n (%) | 185 (17.96) | 16 (11.11) | **0.040** |
| Re-admission, n (%) | 55 (5.34) | 6 (4.17) | 0.54 |
| LOS, n (%) | 17.32±12.05 | 15.81±8.79 | 0.46 |
| MTL30, n (%) | 197 (19.14) | 23 (16.08) | 0.37 |
| 30d Mortality, n (%) | 63 (6.12) | 8 (5.56) | 0.79 |

**Table 3. Pathological results by weekday of surgery.**

| Variable | Weekday | Weekend/Holiday | p-value |
|---|---|---|---|
| | N = 1030 | N = 144 | |
| T-Status, n (%) | | | |
| 0 | 1 (0.10) | 0 (0.00) | |
| 1 | 35 (3.41) | 2 (1.40) | |
| 2 | 57 (5.56) | 5 (3.50) | |
| 3 | 551 (53.70) | 77 (53.85) | |
| 4a | 254 (24.76) | 42 (29.37) | |
| 4b | 128 (12.48) | 17 (11.89) | 0.49 |
| Nodal count | | | |
| Harvest, mean±SD | 24.14±11.84 | 32.79±12.38 | 0.58 |
| Positive, mean±SD | 3.00±5.76 | 3.51±6.08 | **0.0096** |
| N-Status, n (%) | | | |
| 0 | 484 (47.27) | 49 (34.27) | |
| 1 | 288 (27.96) | 51 (35.42) | |
| 2 | 252 (24.47) | 43 (29.96) | **0.013** |
| M-Status, n (%) | | | |
| 0 | 790 (76.85) | 106 (73.61) | |
| 1a | 115 (11.19) | 21 (14.58) | |
| 1b | 123 (11.96) | 17 (11.81) | 0.51 |
| Local R-Status, n (%) | | | |
| 0 | 942 (93.73) | 128 (91.43) | |
| 1 | 44 (4.38) | 11 (7.86) | |
| 2 | 19 (1.89) | 19 (1.89) | 0.14 |
| Grading, n (%) | | | |
| 1 | 46 (4.47) | 10 (7.09) | |
| 2 | 700 (68.90) | 90 (63.83) | |
| 3 | 260 (25.59) | 41 (29.08) | |
| 4 | 10 (0.98) | 0 (0.00) | 0.16 |
| UICC Stage, n (%) | | | |
| I | 74 (7.24) | 6 (4.20) | |
| II | 361 (35.32) | 36 (25.17) | |
| III | 349 (34.15) | 63 (44.06) | |
| IV | 238 (23.29) | 38 (26.57) | **0.020** |

This is particularly relevant as the quality of postoperative care within the first 48 hours after surgery was found to be critical and relevant in terms of patient outcome [15, 16]. In 2011, the National Confidential Enquiry into Patient Outcome and Death report, showed that there had been inadequate post-operative care or monitoring in over 50% of patients who died after emergency surgery in the United Kingdom [17]. In return, studies have been published that suggested improvements in hospital staffing and services might overcome the weekend effect [18, 19]. In German hospitals, providing data to the StuDoQ registry, there is usually a chief resident on duty and an attending trained in colorectal surgery on call. This might explain why neither surgical quality nor postoperative mortality was compromised by weekend surgery, speculating that surgical and postoperative care must be appropriate for these patients.

Interestingly, emergent colorectal cancer resection during the week rather than on the weekend was an independent prognostic factor for the development of major postoperative complications. The higher rate of postoperative complications triggering reoperations can

**Table 4. Multivariable analysis.**

| | | variable | n | OR | CI | p-value |
|---|---|---|---|---|---|---|
| **Colorectal cancer resection** | *30-day mortality* | age | 1174 | 1.04 | 1.01–1.06 | 0.007 |
| | | ASA | 1174 | 3.74 | 2.57–5.44 | <0.001 |
| | | UICC | 1165 | 1.34 | 1.00–1.79 | 0.047 |
| | *MTL30* | ASA | 1172 | 2.51 | 2.03–3.11 | <0.001 |
| | *Clavien-Dindo≥3B* | ASA | 1174 | 1.88 | 1.55–2.28 | <0.001 |
| | | weekday | 1174 | 1.69 | 1.04–2.74 | 0.033 |
| | *Reoperation* | ASA | 1174 | 1.43 | 1.17–1.75 | <0.001 |
| | | weekday | 1174 | 1.76 | 1.02–3.04 | 0.041 |

partially be explained by more extended colectomies and rectal resections performed on a weekday. Although, we found significantly more positive lymph nodes, higher N-stages and ultimately higher UICC tumour stages in patients who underwent emergency colorectal cancer resections on the weekend. About 30% of patients presented with early CRC on the weekend, which supports the results of previously published data describing symptomatic CRC even in UICC stage I and II [20, 21]. When comparing emergent to elective colorectal resections, the latter were shown to have a lower rate of major postoperative complications [8]. This is in line with findings of a study comparing hospital characteristics and the effect on failure to rescue following major surgery. It suggests that microsystem characteristics make the difference in managing postoperative complications, since variabilities between high- and low mortality hospital can only be explained in part by hospital size [22]. Fernandez et al. found no association between day- or night-time urgent colorectal surgery and postoperative morbidities in a retrospective analysis [23]. However, waiting time to surgery and length of hospital stay were significantly longer in patients who underwent urgent colorectal surgery at daytime [23]. This might be associated with a higher postoperative morbidity of these patients [24, 25].

This study inherits some limitations based on the underlying dataset, as hospitals providing data to the StuDoQ registry aim for a certificate from the German Society for General and Visceral Surgery (DGAV). Therefore, the registry data cannot be accounted as a national average but as results acquired in hospitals performing amounts of oncological colorectal resections above the minimum required caseloads of the German Cancer Society (DKG, Deutsche Krebs Gesellschaft) [26]. Thus, mortality after emergency colonic cancer resections remains considerably lower in this study compared to a nationwide analysis based on administrative data [27]. Information regarding the indication to perform emergent resection and out-of-hospital deaths beyond the 30 days period was not available based on the registry data and could therefore not be accounted for. In addition, information about the day of admission was lacking, speculating that weekend surgeries represent the most urgent and worst cases which cannot be postponed to a business day. In turn, this might also explain the small number of patients undergoing emergency colorectal resection at the weekend. However, the use of prospectively collected and annually validated clinical data could be considered a strength, particularly because of the available clinical variables used for adequate risk adjustment in multivariable analysis.

## Conclusion

In this study, no weekend effect was detected among patients who received emergency surgery for colorectal carcinoma in German hospitals. The results of this study clearly state that the surgical quality is not compromised in emergency resections of colorectal cancer in certified

hospitals on the weekend. Future studies need to examine the factors causing a higher rate of major postoperative complications after emergent colorectal resections during the week.

## Supporting information

**S1 Table. Basic patient characteristics.**
(DOCX)

## Acknowledgments

This work has been conducted using the StuDoQ|Colon and Rectal Cancer registry provided by the Study, Documentation and Quality Center (Studien-, Dokumentations- und Qualitäts-zentrum, StuDoQ) of the German Society for General and Visceral Surgery (Deutsche Gesell-schaft für Allgemein- und Viszeral- chirurgie, DGAV) with the ID StuDoQ-2017-0017.

## Author Contributions

**Conceptualization:** Friedrich Anger, Heinz Johannes Buhr, Armin Wiegering.

**Data curation:** Friedrich Anger, Sven Lichthardt, Heinz Johannes Buhr.

**Formal analysis:** Friedrich Anger, Imme Haubitz.

**Methodology:** Imme Haubitz, Armin Wiegering.

**Resources:** Christoph-Thomas Germer.

**Supervision:** Christoph-Thomas Germer, Armin Wiegering.

**Validation:** Johanna Wagner, Stefan Löb.

**Writing – original draft:** Friedrich Anger.

**Writing – review & editing:** Friedrich Anger, Sven Lichthardt, Johanna Wagner, Stefan Löb, Christoph-Thomas Germer, Armin Wiegering.

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
