## [Decision Letter · Decision Letter 0]

10 Sep 2022

PONE-D-22-11972Is there a weekend effect in emergency surgery for colorectal carcinoma? Analysis from the German StuDoQ registryPLOS ONE

Dear Dr. Anger,

Thank you for submitting your manuscript to PLOS ONE. After careful consideration, we feel that it has merit but does not fully meet PLOS ONE’s publication criteria as it currently stands. Therefore, we invite you to submit a revised version of the manuscript that addresses the points raised during the review process.

We look forward to receiving your revised manuscript.

Kind regards,

Leonardo Solaini, MD

Academic Editor

PLOS ONE

Journal Requirements:

Reviewers' comments:

Reviewer's Responses to Questions

**Comments to the Author**

Review Comments to the Author

Reviewer #1: The topic chosen by the authors is no doubt interesting.

The so-called weekend effect is a well-studied but yet-to-be-demonstrated and defined hypothetical impact suggesting that patients hospitalized and treated on weekends may have different outcomes than those admitted and treated during normal workday hours.

The authors immediately focus in the Introduction one of the major problems in the interpretation of this effect “The published data originates from different health care systems varying in the provision of acute and hospital care, weakening their comparability “. Choosing to select a defined disease confined to a homogeneous health system can be a very effective choice. The aims of the study are clearly stated. They also provide an opportunity to learn about a specific tool of the German Health care system, MTL 30, well explained in the bibliographic references provided.

Both the structure of the statistical analysis and the definition of primary and secondary outcomes are well defined.

The conclusion of the study is consistent with the available data and should be an incentive for surgeons to require in their institutions a work structure that considers the skills of the medical teams that are assembled.

An adequate arrangement of resources protects the patient (and also the surgeons) any day of the week.

Some notes

In Results: the conversion rate in open surgery should be specified in the text

In tab 1 and 2 it is clear that in all cases an anastomosis was performed. In table 2 the rate of diverting stoma is specified; it would be interesting to know the indications for protective stomas.

In the tab 2 the part related to ostomy should be better clarified: prior? After resection surgery: Is this a redo surgery due to complication? .

Unfortunately, due to the characteristics of the database, it is not possible to know how many patients were operated for occlusion, how many for perforation and / or bleeding nor the data of mortality at 90 days and this leads to lose part of the value of the paper.

Reviewer #2: This is a generally well written paper on an important and topical subject. The authors have obtained data on a large number of patients with CRC treated as an emergency. The outcome measures chosen are relevant and well recognized in similar studies. Their data was obtained form the German StuDoQ registry, which would be familiar to German readers. However, for the benefit of non-German readers it would be helpful if the authors could add a sentence or two in the first paragraph of the Methods section to indicate if this registry captures all cases of CRC in Germany, or if it is selective.

There is one area of the results/discussion where I would like further clarification- Overall there was a higher rate of severe post operative complications and return to theatre for weekday versus weekend patients, but there was no difference in mortality or other outcomes. The reason for this may be due to case mix. The authors allude to the fact that defunctioning stoma is widely used in Germany for emergency cases. It would be helpful if the authors could explicitly state (or show on a table) whether there is any significant difference in the type of surgery performed on different days. For example were there more resectional operations done on Mon-Fri and more defunctioning operations done on a weekend?

There are also several areas where the grammar/ syntax should be changed-

Methods Section - "Study Population" - line 3-4 should be "classified as an emergency"

Methods Section - "Study Population" - Line 9 - should be "clinical features such as the American....."

Methods Section - "Study Population" - Line 13 - should be "logistic measures such as the day..."

Methods Section - "Statistical analysis and outcome measures" - Line 1- should be " a two-step approach"

Results Section - "Emergency Surgery" - Line 2 - should be "where surgery had been performed on a week day rather"

Results Section - "Emergency Surgery" - penultimate line - should be "The timing of emergency surgery...."

Discussion Section - Paragraph 2 - Line 10 - should be "as bridge to surgery is the preferred rout in German hospitals,"

Discussion Section - Paragraph 2 - Line 11- should be "multimodality treatment prior to resection."

Discussion Section - Paragraph 3 - Line 4-6 - The sentence should be rephrased so that it reads as - "In 2011, the National Confidential Enquiry into Patient Outcome and Death report (17), showed that there had been inadequate post-operative care or monitoring in over 50% of patients who died after emergency visceral surgery in the United Kingdom."

---

## [Author Response · Author response to Decision Letter 0]

27 Sep 2022

We would like to thank the reviewers for their valuable time and useful contributions. We highly appreciate the inputs you have given which definitely helped to improve our manuscript. In the following, please find point-to-point answers to your comments. All corrections or implementations are highlighted in the revised manuscript.

Reviewer #1:

1) In Results: The conversion rate in Open surgery should be specified in the text.

Thank you for this important point. We added the significant difference in the surgical approach to the results section in the text.

2) In tab 1 and 2 it is clear that in all cases an anastomosis was performed. In table 2 (now table 1) the rate of diverting stoma is specified; it would be interesting to know the indications for protective stomas.

Thank you for this query. Unfortunately, the StuDoQ registry does not provide data on the indications for protective stomas. But the most reasonable indications for diverting stoma prior surgery is a tumor stenosis, simultaneously to resection surgery is protection of the actual anastomosis. In case of a diverting stoma placement after resection surgery, patients underwent re-operation due to anastomotic leakage. 

3) In the tab 2 the part related to ostomy should be better clarified: prior? After resection surgery: Is this a redo surgery due to complication?

Thank you for this valid point. We restructured this part in table 2 (now table 1) to clarify this information for the reader. It now says whether patients received a diverting stoma and at what time with regard to the resection surgery. Although the registry does not provide information on the indication for diverting stoma placement, all patients who received a diverting stoma after resection surgery had re-operation performed. 

Reviewer #2:

1) There is one area of the results/discussion where I would like further clarification- Overall there was a higher rate of severe post operative complications and return to theatre for weekday versus weekend patients, but there was no difference in mortality or other outcomes. The reason for this may be due to case mix. The authors allude to the fact that defunctioning stoma is widely used in Germany for emergency cases. 

We appreciate this relevant comment. Indeed, severe post-operative complications and return to theatre for weekday patients occurred (significantly) more often, but the same was found for re-admission, length of stay, MTL30 and 30d mortality. The latter remain without statistical significance, most likely due to the rather small sample size of weekend surgeries included. The higher rate of postoperative complications triggering reoperations can be explained by more extended colectomies and rectal resections performed on a weekday. This might be due to the fact that in case of obstructing colorectal cancer the sole creation of an ostomy without resection surgery is performed more often on a weekend. However, patients without resection of their colorectal cancer were not included in this study. Case mix seems to be a rather unlikely reason for a higher rate of postoperative complications and reoperations for weekday patients, since UICC tumor stages in patients who underwent emergency colorectal cancer resections on the weekend were significantly higher than in patients who received emergency surgery on a weekday. In addition, no difference was found for Age, BMI, ASA, functional status and comorbidities between the two groups (new table 1).

2) It would be helpful if the authors could explicitly state (or show on a table) whether there is any significant difference in the type of surgery performed on different days. For example were there more resectional operations done on Mon-Fri and more defunctioning operations done on a weekend?

Thank you for this important query. There is a significant difference in resection strategy between weekday and weekend surgery, mostly due to more rectum resections on weekdays (10.4%) compared to weekends (2.8%) (new table 1). This led to the conclusion, that emergencies due to e.g. obstructing rectal cancer were managed by decompressing ostomy as bridge to surgery, as pointed out in the discussion section (second paragraph). Cases with no resection but just decompressing or defunctioning surgery in terms of stoma creation were not included in this study. Regarding the creation of a diverting stoma, no difference could be found between weekday and weekend emergency resection surgeries due to colorectal cancer. 

3) There are also several areas where the grammar/ syntax should be changed-

Please excuse us for this shortcoming and thank you for the detailed revision. We changed all errors according to your suggestions.

---

## [Editor Report · Decision Letter 1]

19 Oct 2022

Is there a weekend effect in emergency surgery for colorectal carcinoma?

 Analysis from the German StuDoQ registry

PONE-D-22-11972R1

Dear Dr. Anger,

We’re pleased to inform you that your manuscript has been judged scientifically suitable for publication and will be formally accepted for publication once it meets all outstanding technical requirements.

Kind regards,

Leonardo Solaini, MD

Academic Editor

PLOS ONE

---

## [Editor Report · Acceptance letter]

25 Oct 2022

PONE-D-22-11972R1 

Is there a Weekend Effect in Emergency Surgery for Colorectal Carcinoma?
 Analysis from the German StuDoQ Registry 

Dear Dr. Anger:

I'm pleased to inform you that your manuscript has been deemed suitable for publication in PLOS ONE. Congratulations! Your manuscript is now with our production department. 

Kind regards, 

on behalf of

Dr. Leonardo Solaini 

Academic Editor

PLOS ONE